# Am I truly monolingual? Exploring foreign language experiences in monolinguals

**Sofía Castro** [1]*, **Zofia Wodniecka**[1], **Kalinka Timmer**[1,2]

**1** Psychology of Language and Bilingualism Lab, Institute of Psychology, Jagiellonian University, Kraków, Poland, **2** Faculty of Psychology, University of Warsaw, Warsaw, Poland

* sofia.gonzalez.castro@doctoral.uj.edu.pl

## Abstract

Monolingualism has typically been understood as a homogeneous phenomenon. The linguistic experiences of monolinguals are usually overlooked when analysing the impact of foreign language experiences on language processing and cognitive functioning. In this study, we analyse the linguistic experiences of 962 English-speaking individuals from the United Kingdom (UK) who identified as monolinguals. Through an online survey, we found that more than 80% of these monolinguals had learned at least one foreign language, dialect, or type of jargon. More than half of this 80% of monolinguals also used languages they had learned at some point in their lives. Moreover, nearly 40% of all the studied monolinguals confirmed that they had been passively exposed to foreign languages or dialects in their environment; approximately a fourth of these monolinguals who declared exposure to at least one foreign language (or dialect) confirmed that they also used these languages. Furthermore, activities that involved passive use of languages (i.e., activities that require reading or listening but do not require speaking or writing; e.g., watching TV) were occasionally carried out in foreign languages: around 26% of these monolinguals confirmed the passive use of more than one language. Lastly, around 58% of monolinguals who had visited one or more non-English-speaking countries declared the active use of foreign languages during their stay(s). These results suggest that the linguistic experiences of monolinguals from the UK often include exposure to and use of foreign languages. Moreover, these results show the need to consider the specificity of the monolingual language experience when analysing the impact of foreign languages on cognitive functioning, as differences in the language experiences of bilinguals also have divergent impacts on cognition. Lastly, monolingual experiences are different from bilingual experiences; therefore, existing questionnaires that evaluate language experiences should be adapted to capture the particular linguistic experiences of monolinguals.

## Introduction

Bilingualism (and multilingualism) is a common phenomenon in society. Millions of individuals around the world know and use more than one language on a daily basis. In contrast to

**Data Availability Statement:** The complete filtered and organized dataset used for the analysis is freely available at: https://osf.io/ps2k6/.

**Funding:** SC and ZW received funding from the European Union's Horizon 2020 research and

innovation programme under the Marie Skłodowska Curie grant agreement No 765556 - The Multilingual Mind. KT was supported by funding from Narodowa Agencja Wymiany Akademickiej in Poland with an Ulam grant (PPN/ULM/2019/1/00215). The open-access article processing charge (APC) has been funded by a grant awarded to KT within the Programme "Excellence Initiative - Research University" of Warsaw University. The funders had no influence in the study design, data collection and analysis, decision to publish, or preparation of the manuscript.

**Competing interests:** The authors have declared that no competing interests exist.

more traditional and restrictive definitions [1], current definitions of bilingualism emphasise the use of more than one language rather than proficiency [2, 3]. Considering the bilingual language experience as heterogeneous and dynamic complicates its conceptualization, and a variety of labels, definitions, and descriptions are used by the scientific community (for a review of the conceptualization of bilingualism in the 21ˢᵗ century, see [4]). Efforts devoted to the conceptualization and description of bilingualism go hand in hand with the emergence of research investigating the impact of different long-term (e.g., [5]) and short-term bilingual experiences (e.g., [6–8]) on linguistic and non-linguistic cognitive processes (for a review, see, e.g., [9]). While this literature takes into consideration the heterogeneity of bilingualism and how different bilingual language experiences modulate different cognitive processes, monolinguals are still seen as a homogeneous group. However, is monolingualism truly a homogeneous phenomenon? The conceptualization and description of monolingual language experiences [10] and their impact on language and cognition (e.g., [11]) have been largely overlooked. One example of oversimplification in the perception of monolingualism was demonstrated in a recent study by Nichols et al. [12], where the grouping of individuals as monolinguals or bilinguals was based on one or two questions (1 –"How many languages do you speak? Select 1–20"; 2 –"What language(s) do you primarily speak at home?").

Researchers agree that understanding the heterogeneity of bilingual language experiences is crucial to comprehend how bilingualism shapes cognition. Several studies have suggested that bilingualism is associated with more efficient domain-general cognitive functions (for reviews, see e.g., [13–16]). Nevertheless, the idea that bilinguals, compared to monolinguals, have enhanced domain-general functioning is still heavily debated (e.g., [17, 18]). Recent research has proposed that this modulation may be due to differences in how bilinguals use their languages on a daily basis (e.g., [5, 19, 20]). However, to fully understand the relation between bilingualism and cognition, it is also essential to ask whether the pattern of language use in monolinguals is truly homogenous and how monolingual language experiences might impact cognitive processes. The current paper is a first attempt at evaluating the assumption of homogeneity in monolinguals by exploring their foreign linguistic experiences.

## Monolingual language experiences and its impact on language and cognition

Studies with monolinguals have found that short-term foreign language exposure as well as foreign language learning influence both language processing (e.g., [21–24]) and executive control in monolingual children [25] and monolingual adults [26]. For example, Kurkela et al. [23] found that adult Finnish monolinguals who were passively exposed to Mandarin speech sounds for four days showed greater neural discrimination to novel verbal and nonverbal auditory stimuli. Furthermore, Bice and Kroll [21] discovered that monolinguals immersed in linguistically diverse environments seemed to develop higher sensitivity to non-native phonological contrasts when learning words in a new language. Exposure to foreign languages in the direct environment (e.g., the presence of another language in the neighbourhood) is sometimes referred to as "ambient exposure" [27]. However, researchers investigating ambient exposure do not always differentiate between the mere presence of a foreign language in the environment and the degree of its use [28, 29]. Regarding language learning and executive control, Sullivan et al. [26] found that 6 months of foreign language learning modulates the electrophysiological responses of monolinguals in linguistic and non-linguistic conflict tasks. Even shorter learning periods have an effect: there is evidence that young monolingual children's performance improves on a battery of executive control tasks after a 20-day language learning period [25]. These studies suggest that the impact of foreign languages on linguistic

and cognitive processes can occur at the early stages of language learning and even without active language use.

The studies discussed above provide the first evidence that monolingual participants do not always have similar or unchangeable linguistic experiences [10, 11, 21]. For example, monolinguals living in a metropolis might be immersed in a more diverse context compared to monolinguals living in a small town; monolinguals from different age groups might have learned foreign languages in and outside school to a different extent; some monolinguals might have spent part of their lives abroad, whereas others might not have travelled at all. However, studies comparing monolinguals and bilinguals tend to consider only the heterogeneity of the linguistic experiences of bilingual participants because these studies assume that monolinguals either had no experiences with foreign languages or the experiences they might have had are anecdotical and of no relevance. This lack of attention might have affected the way in which we understand the relationships between bilingualism, language, and cognition. In conclusion, more attention needs to be devoted to understanding the extent to which monolinguals differ in their foreign language experiences.

## Assessing language experiences in bilinguals and monolinguals

Bilingual language experiences are usually measured by collecting information on the linguistic profiles of participants, such as their proficiency in different languages, their language history, and their communicative patterns. Language proficiency is often defined with objective measures (i.e., tests, e.g., PPVT-IV [30], MINT [31], BNT [32], LexTALE [33], EVT-3 [34]), but many questionnaires also include assessments of subjective proficiency, together with different measures of language history, language use, and language experience (e.g., Code-Switching and Interactional Context Questionnaire [5], LSBQ [35], LHQ3 [36], LEAP-Q [37], BSWQ [38]).

However, questionnaires that evaluate participants' language experiences and language use focus on bilinguals, neglecting the different and specific linguistic experiences of monolinguals. Participants categorized as monolinguals are either not asked to fill out more detailed questions about their language experience, or the information on their linguistic experiences is gathered using questionnaires developed for bilinguals (e.g., [39]). Unfortunately, these questionnaires do not capture crucial aspects of the monolingual language experience, such as passive exposure to foreign languages and passive language use. These aspects are often overlooked in these questionnaires because they tend to be less important for the study of bilingual language experiences. By passive exposure to foreign languages, we refer to situations in which foreign languages are present in the environment but the individual does not interact with these languages in any manner (e.g., by working with people who speak a different language and who often communicate between themselves in that language; by living in a neighbourhood with a high number of foreign language speakers). By passive use of languages, we refer to the intentional use of foreign languages in activities that involve some reading or listening but do not require speaking or writing. In conclusion, the linguistic diversity of a monolingual environment (e.g., (in)formal language learning and passive exposure to other languages, or sporadic active use of foreign languages) tends to be overlooked.

## The present study

To better understand the diversity in the linguistic profiles of individuals who categorize themselves as monolinguals, we made a first attempt at describing the linguistic experiences of monolinguals by asking native English speakers from the United Kingdom (UK) to complete a short questionnaire about their language experiences.

We focused on monolinguals from the UK because, according to the 386 Special Eurobarometer [40], (1) UK citizens are less likely to speak foreign languages compared to most other citizens from Europe (except for Italy and Hungary); and (2) they are also less likely to have learned a foreign language (only surpassed by Portugal and Spain). Moreover, English is the most common language that is spoken as a foreign language in the European Union [40]. Thus, native English speakers from the UK might have the smallest chance of encountering foreign languages when communicating with other people. For these reasons, monolinguals from the UK are expected to have less experience with other languages, in terms of both exposure and use.

In this analysis, we focus on four aspects that we believe are relevant when exploring monolingual language experiences: foreign language learning, passive exposure to foreign languages in the home country (i.e., UK), linguistic experiences while visiting/residing in non-English-speaking countries, and passive use of languages (native and non-native languages). Finding heterogeneity in the linguistic profiles of individuals who classify themselves as monolingual would support the need to consider the variability of "monolingual" language experiences in future research.

## Method

### Participants

A total of 970 monolinguals aged between 18 and 82 years old ($M = 38.40$, $SD = 13.32$; 556 females, 413 males, 1 not reported) were recruited using Prolific (www.prolific.co). The data were collected as part of a larger study comparing cognitive processes in bilinguals and monolinguals, the results of which are beyond the scope of the present paper. All participants gave informed written consent before starting the survey and received financial compensation for their participation. The experiment met the requirements of the Ethics Committee of the Institute of Psychology of Jagiellonian University concerning experimental studies with human subjects.

To delimit the recruitment process to only monolinguals, we applied four pre-screening filters provided by Prolific: we only invited participants who were born in the UK, who were currently living in the UK, who reported knowledge of only English, and who were raised with only one language (see S1 Appendix for detailed information on the pre-screening questions). According to their Prolific information, most participants identified as "Caucasian" or "White/Caucasian" (891 participants or 91.86%), had British nationality (967 or 99.69%), and spent most of their childhood and adolescence in the UK (961 or 99.07%). Participants were of average subjective socioeconomic status (SES) ($M = 5.42$, $SD = 1.55$), reported by placing themselves on a scale from 1 (lowest SES) to 10 (highest SES); most (74.54%) were working full- or part-time.

### Procedure

Participants completed an online survey that lasted approximately 5 minutes. We asked them to provide information regarding (1) learning of foreign languages/dialects/types of jargon; (2) passive exposure to foreign languages/dialects in the UK; (3) passive use of languages/dialects; and (4) past stays in non-English-speaking countries. As we wanted to include questions that tapped into the specific linguistic experiences of monolinguals, the survey we created went beyond the typically used questionnaires. Still, to be able to evaluate monolinguals' level of proficiency in the languages they learned, we included the widely-used approach of separately self-assessing each domain of language on a numeric scale (i.e., reading, listening, writing, speaking). In addition, we also asked them to report the age of acquisition of each language.

For the sake of simplicity, in the manuscript, the term "language" includes dialect and jargon when referring to learning foreign languages; it only includes dialect when referring to exposure to foreign languages, passive use of languages, and stays abroad.

First, participants who confirmed that they had learned foreign languages were also asked to report the names of the languages and for how long they had learned them; they were also asked to confirm whether they ever used them for any reason. Second, participants who indicated exposure to foreign languages in the UK also included the names of these languages, the length of their exposure to them, as well as whether they ever used these languages. Third, regarding the passive use of languages, participants indicated the percentage of daily passive use of their languages (e.g., while watching TV or listening to the radio), together with the names of these languages. We use the term "passive use of languages" instead of "use of languages for reading or writing" to prevent participants associating these activities with high language proficiency, as not all activities that involve passive language use necessarily require high proficiency (e.g., browsing the internet, listening to music). Creating the association between passive use and high proficiency could prevent participants from reporting all the languages they passively used as they considered themselves proficient only in English. Moreover, participants were asked to indicate whether the COVID-19 pandemic had impacted how they passively use their languages. Participants who reported a change as a result of the COVID-19 pandemic had to indicate their passive use of languages both before and during the COVID-19 pandemic. Fourth, participants who declared having stayed abroad in non-English-speaking countries also indicated the names of the countries, the lengths of their visits, the languages spoken in these countries, whether they used foreign languages during their stay, whether the languages from those countries were present in their current environment, and whether they used these languages after they returned to the UK (see S2 Appendix for the complete survey).

## Data cleaning

**Data exclusion.**   Five participants were removed for the following reasons: technical errors (1 participant), inconsistencies in their Prolific information (1 participant), missing data (2 participants), or low self-reported English proficiency (1 participant who had an average proficiency of 1.5 out of 10 was excluded).

Although all 970 participants were native speakers of English, nine reported that they had learned English as a foreign language. Although they reported that English was a foreign language to them, these nine participants indicated high proficiency in it ($M = 9.81$, $SD = 0.39$) and acquisition of English from birth ($M = 0.11$ years, $SD = 0.33$). Hence, we can presume that these participants were indeed native speakers of English. However, out of these nine, three participants indicated the use (present or past) of at least one of the foreign languages learned. Since these three participants' lists of the foreign languages they had learned also included their native English, it was impossible to know whether they included English in the foreign languages they listed when answering the question about the use of foreign languages. Taking this into consideration, these three participants were removed. The remaining six participants were kept in the analysis after removing English from the list of foreign languages learned. Hence, data from 962 participants were analysed.

**Data classification.**   A total of 61 participants (6.34% of the total 962) indicated a change in their passive use of languages as a consequence of the COVID-19 pandemic. However, out of these 61 participants, 18 described identical passive use of languages before and during the COVID-19 pandemic (e.g., 100% of passive use of English before the COVID-19 pandemic and 100% of passive use of English during the COVID-19 pandemic). Therefore, these 18 participants were treated as if the COVID-19 pandemic had no impact on their passive use of

languages. For the remaining 53 participants who declared an impact of COVID-19 in their passive use of languages and indeed reported different passive language use before and during COVID-19, we observed no significant differences in their passive use of languages before and during the COVID-19 pandemic, $t(52) = 0.68$, $p = .501$. For these 53 participants, the data corresponding to their language use during the COVID-19 pandemic were used as these are the most recent reports. To sum up, the analysis of the passive use of languages includes data from all 962 participants, regardless of whether COVID-19 impacted or not their passive use of languages.

When asked about their stays in non-English-speaking countries, one participant reported two identical stays (i.e., two stays in the same country for the same amount of time) but assigned a different foreign language to each of these two identical stays. Therefore, we consider that this participant had been abroad once and was exposed to two different languages during their visit.

**Organization of the data.** First, we calculated the average proficiency from the self-reported proficiency in reading, listening, writing, and speaking for each language that participants reported having learned. The language with the highest average proficiency was coded as their second language (L2), followed by the third language (L3) and the fourth language (L4). Second, the languages to which participants were exposed in the UK were reordered by length of exposure. Third, languages passively used by participants were reordered according to the percentage of passive use. Fourth, the non-English-speaking countries in which participants had stayed were reordered according to the length of their stay.

Scottish Gaelic and Scots Gaelic were recoded as Gaelic; Northern Irish was recoded as Irish. Chinese Mandarin was recoded as Mandarin; "American", "English USA", and "English with subtitles" were recoded as English; Holland was recoded as the Netherlands.

The complete filtered and organized dataset used for the analysis is freely available at: https://osf.io/ps2k6/?view_only=31228c2b0e93414392d39fe2734f043a.

## Results

### Learning languages

Most participants (83.26%) reported learning a foreign language at some point in their lives; more than half of participants who had learned at least one foreign language indicated that they had used one or more of these languages (53.43%; See Fig 1). The most common languages reported were French, German, and Spanish. Furthermore, participants who reported having learned one or more foreign languages were significantly younger ($M = 37.97$ years, $SD = 12.91$) than participants who had not learned foreign languages ($M = 40.68$ years, $SD = 14.71$); Welch t-test, $t(212.36) = 2.17$, $p = .031$.

Participants reported the acquisition of English from birth and the acquisition of foreign languages throughout their life, though most often during adolescence. For participants who had learned at least a second language, the average self-rated proficiency in their native English was higher compared to their average proficiency in L2 ($M_{English} = 9.89$; $M_{L2} = 2.66$), $t(800) = 121.87$, $p < .001$; the difference in age of acquisition between English and L2 was also significant ($M_{English} = 0.02$ years; $M_{L2} = 13.12$ years), $t(800) = -62.71$, $p < .001$. Participants who had learned at least two foreign languages were more proficient in their L2 compared to their L3 ($M_{L2} = 2.94$; $M_{L3} = 1.73$), $t(477) = 22.05$, $p < .001$. In addition, their L2 (vs. L3) was acquired at a younger age ($M_{L2} = 13.21$ years; $M_{L3} = 14.88$ years), $t(477) = -3.56$, $p < .001$, and their L2 (vs. L3) had been learned for longer ($M_{L2} = 4.28$ years; $M_{L3} = 2.56$ years), $t(477) = 10.84$, $p < .001$. Participants who had learned three foreign languages were more proficient in their L3 than in their L4 ($M_{L3} = 2.19$; $M_{L4} = 1.26$), $t(177) = 13.78$, $p < .001$; they had learned their L3 (vs. L4)

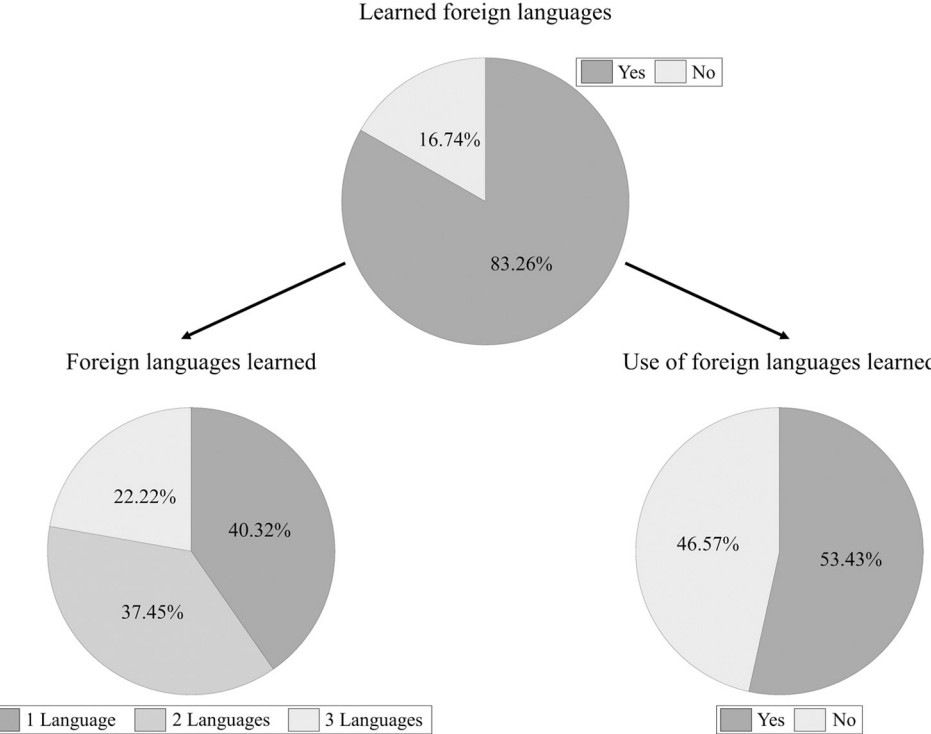

**Fig 1. Relative frequency of foreign language learning.** The pie chart above represents the proportion of participants who reported having learned (n = 801) or not learned foreign languages (n = 161). Out of the total number of participants who had learned foreign languages, the lower-left pie chart shows the proportion who had learned one (n = 323), two (n = 300), or three (n = 178) foreign languages; the lower-right pie chart shows the proportion who reported use (n = 428) or no use (n = 373) of the foreign languages learned.

for longer ($M_{L3}$ = 2.81 years; $M_{L4}$ = 1.77 years), $t(177)$ = 5.91, $p < .001$, but they did not differ in the age of acquisition of their L3 and L4 ($M_{L3}$ = 15.89 years; $M_{L4}$ = 16.89 years), $t(177)$ = -1.07, $p$ = .286. See Table 1 for detailed information regarding the proficiency, age of acquisition, and time spent learning foreign languages, as well as for information on English proficiency, age of acquisition, and the age at which English fluency was reached. The list of languages reported by participants can be consulted in S3 Table.

## Exposure to foreign languages in the UK

Of the 962 participants, 39.09% confirmed having been exposed to foreign languages in the UK at some point in their lives: 24.47% of them reported the use of at least one of these

**Table 1. Means (and standard deviations) for self-rated proficiency, age of acquisition, and time learning L2, L3, and L4, as well as self-rated English proficiency, age of acquisition of English, and age at which English fluency was reached.**

| | L1 English (n = 962) | | L2 (n = 801) | | L3 (n = 478) | | L4 (n = 178) | |
|---|---|---|---|---|---|---|---|---|
| | *M* (SD) | Range | *M* (SD) | Range | *M* (SD) | Range | *M* (SD) | Range |
| Self-rated proficiency [a] | 9.87 (0.54) | 5–10 | 2.66 (1.62) | 0–10 | 1.73 (1.31) | 0–6.5 | 1.26 (1.07) | 0–5.25 |
| Age of acquisition [b] | 0.04 (0.40) | 0–6 | 13.12 (5.91) | 0–54 | 14.88 (7.96) | 4–64 | 16.89 (9.07) | 4–57 |
| Time learning [b] | | | 3.91 (2.85) | 0.083–50 | 2.56 (1.70) | 0–12 | 1.77 (1.60) | 0.083–8 |
| Age of acquired fluency [b] | 2.10 (1.21) | 0–12 | | | | | | |

[a] Proficiency is measured on a scale from 0 to 10, with 10 being the highest proficiency.

[b] Age of acquisition, time spent learning, and age at which English fluency was reached are measured in years.

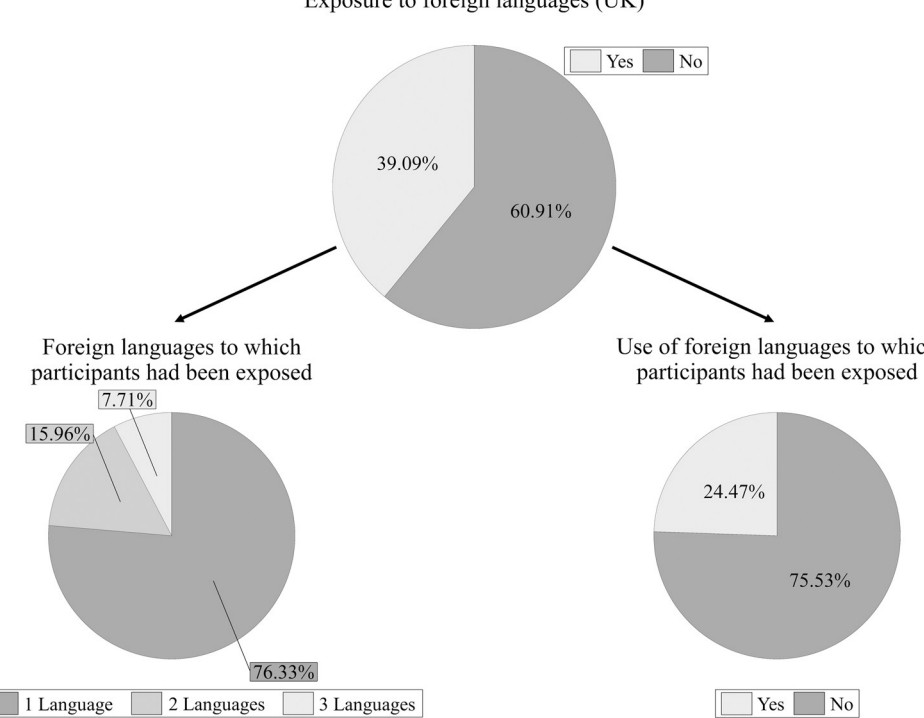

**Fig 2. Relative frequency of foreign language exposure in the UK.** The pie chart above represents the proportion of participants who reported having been exposed (n = 376) or not exposed (n = 586) to foreign languages in the UK. Out of the total number of participants who had been exposed to foreign languages, the lower-left pie chart shows the proportion of participants who had been exposed to one (n = 287), two (n = 60), or three (n = 29) foreign languages; the lower-right pie chart shows the proportion of participants who reported the use (n = 92) or no use (n = 284) of the foreign languages to which they had been exposed.

languages (see Fig 2). Participants were exposed to foreign languages for between 4 and 6 years on average, with a lot of variability among participants (see Table 2 for detailed information on the length of exposure). The list of foreign languages to which participants had been exposed is presented in S2 Table.

## Passive use of languages

Approximately one fourth of participants (26.30%) indicated the passive use of more than one language (See Fig 3). However, most of the activities that involved passive use of languages (e.g., listening to music) were carried out in their native English. More specifically, 908

**Table 2. Average exposure (and standard deviations) and range of exposure to foreign languages in the UK.**

| Language | Average exposure (years) | Range (years) |
|---|---|---|
| Foreign language 1 (n = 376) | 6.03 (9.17) | 0.083–52 |
| Foreign language 2 (n = 89) | 4.91 (7.31) | 0.083–45 |
| Foreign language 3 (n = 29) | 6.12 (8.47) | 0.083–35 |

Note. A total of 376 participants were exposed to at least one foreign language (i.e., including participants who were exposed to more than one foreign language); 89 participants were exposed to at least two foreign languages; and 29 participants were exposed to three foreign languages. The languages were categorized based on the exposure time: Foreign Language 1 represents the language to which participants had been exposed for longest, followed by Foreign Language 2, Foreign Language 3, and Foreign Language 4.

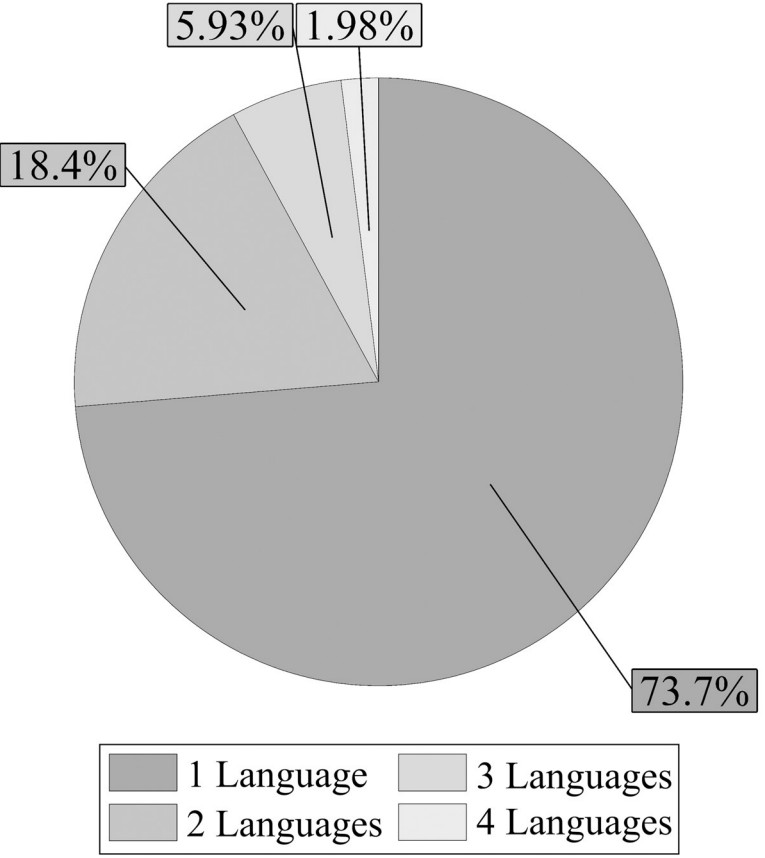

**Fig 3. Relative frequency of passively used languages (native and foreign).** A total of 709 participants reported the passive use of one language; 177 reported the passive use of two; 57 the passive use of three; and 19 reported the passive use of four languages.

participants (94.39%) used English for a minimum of 90% of the time, while 32 participants (3.33%) reported passive use of English for 50% to 87% of the time. Surprisingly, 21 participants (2.18%) did not report passive use of English in any situation; one participant (0.10%) passively used English less frequently than other languages. The list of languages reported by participants is shown in S3 Table.

## Linguistic experiences in non-English-speaking countries

Only 9.25% of participants had stayed in one or more non-English-speaking countries. From the total number of participants who had been abroad, 58.43% indicated that they had used foreign languages during their stays (see Fig 4); 7.87% (out of the total 89 participants who had stayed abroad) reported the current presence of those foreign languages in their environment; 7.87% (out of the total 89) reported the current use of those languages. Detailed information about the length of the stays abroad can be found in Table 3. The list of countries reported by participants can be found in S4 Table.

## Summary

These results show that more than 80% of participants had learned one or more foreign languages, and more than half of those had used at least one of their foreign languages at some

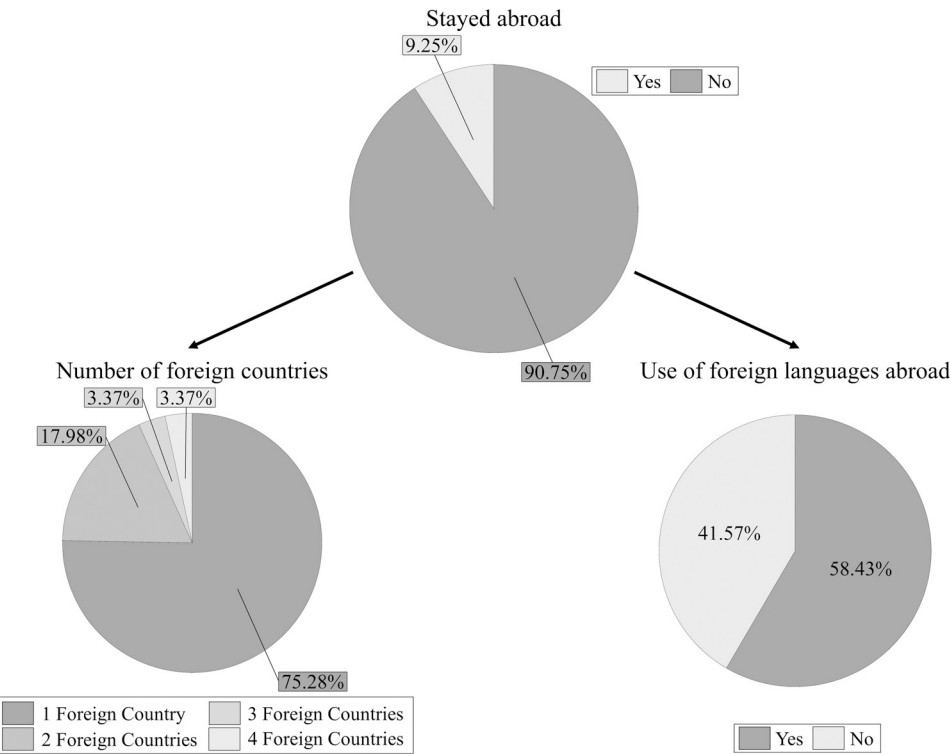

**Fig 4. Percentage of monolinguals who had stayed abroad in non-English-speaking countries.** The top pie chart represents the proportion of participants who reported that they had (n = 89) or had not stayed abroad (n = 873) in non-English-speaking countries. Out of the total number of participants who had been abroad, the lower-left pie chart shows the proportion of participants who had stayed in one (n = 67), two (n = 16), three (n = 3), or four (n = 3) foreign countries; the lower-right pie chart shows the proportion of participants who reported using (n = 52) or not using (n = 37) foreign languages during their stays abroad.

point in their lives (around 53%; see Fig 1). On average, participants had learned (a) foreign language(s) for one and a half to four years, usually starting during adolescence. However, the average level of proficiency in their foreign languages was generally low (L2: $M = 2.66$, $SD = 1.62$; L3: $M = 1.73$, $SD = 1.31$; L4: $M = 1.26$, $SD = 1.07$) compared to English ($M = 9.87$, $SD = 0.54$). Those who had learned one or more foreign languages were significantly younger

**Table 3. Mean (and standard deviations) length of stay and range of length of stay in non-English-speaking countries.**

| Country | Average length of stay (years) | Range (years) |
|---|---|---|
| Foreign country 1 (n = 89) | 2.56 (4.38) | 0.083–25 |
| Foreign country 2 (n = 22) | 1.10 (1.21) | 0.083–4 |
| Foreign country 3 (n = 6) | 0.46 (0.30) | 0.167–1 |
| Foreign country 4 (n = 3) | 0.31 (0.17) | 0.167–0.5 |

Note. A total of 89 participants had stayed at least in one non-English speaking country (including participants who had stayed in more than one foreign country); 22 participants had stayed in at least two non-English speaking countries; 6 had stayed in at least three non-English speaking countries; and 3 had stayed in four non-English speaking countries. The countries were categorized based on the length of stay: Foreign Country 1 is the country in which a given participant had spent most time, followed by Foreign Country 2, Foreign Country 3, and Foreign Country 4.

than those who had not learned any foreign language. In addition to learning languages, participants had also been exposed to foreign languages in the UK: almost 40% of the total of 962 participants reported present or past exposure to one or more foreign languages; a fourth of this 40% also declared use of the languages to which they had been exposed (around 24%; see Fig 2).

With respect to their passive use of languages, around 26% of participants reported the passive use of more than one language (see Fig 3). Nevertheless, most activities that involved passive use of languages were carried out in English: almost 95% of the total 962 participants used English passively for at least 90% of the time. Surprisingly, a few participants indicated no passive use of English or less passive use of English compared to other languages.

Regarding linguistic experiences abroad, only about 9% of participants had stayed in one or more non-English-speaking countries (see Fig 4). Still, more than half of them reported having used one or more foreign languages during their stay(s) abroad. In general, they stopped using these foreign languages when they went back to the UK, although a small number of participants who had stayed abroad reported current use of those languages (approximately 8%). In conclusion, these results indicate surprisingly rich linguistic experiences among individuals who declare themselves to be monolingual.

## Discussion

In this study, we explored the assumption that individuals who consider themselves monolingual are homogenous. To this end, we analysed and described the linguistic and communicative experiences of a group of monolinguals from the UK. Our results indicate that monolinguals have a diverse linguistic profile that includes frequent foreign language learning (formal and/or informal) as well as past and present exposure and use of foreign languages. This reveals that monolinguals actually have heterogenous rather than homogenous language experiences. Therefore, monolingual language experiences should be considered in more detail when analysing the impact of foreign languages on linguistic and cognitive processes (e.g., see [10, 11, 21]).

### Heterogeneous monolingual language experiences

Most participants who completed our survey (approximately 80%) had learned one or more foreign languages for an extended period. Moreover, half of these participants had also used one or more foreign languages at some point in their lives (see Fig 1). This suggests that their foreign language knowledge was sufficient to allow successful communicative exchanges. Nevertheless, their subjective proficiency in the foreign languages they had learned was low on average (around 2 on a scale from 0 to 10). These two findings might seem contradictory at first, but self-rated measures of proficiency are sometimes criticised due to their imprecision [41] and the fact that they might also be subject to biases [42]. Despite their subjective nature (self-report), our data indicate that monolinguals who had learned foreign languages used them frequently. We also observed that participants' age was linked to language learning experiences, with more frequent foreign language learning among younger participants. This effect is probably related to the increased importance given to foreign languages in education [43] and suggests that monolinguals will more often have experiences with foreign languages in the future.

Notably, the use of languages did not only occur in the context of language learning. Approximately one fourth of participants who declared present or past exposure to foreign languages also reported the use of these foreign languages in other situations (see Fig 2). Moreover, one fourth of participants passively used more than one language daily (see Fig 3),

although the predominant language was English. Even though the passive use of a language does not imply that this language is fully understood, it indicates intentional involvement with foreign languages. Unfortunately, participants were not asked to specify during which activities they were passively exposed to foreign languages or passively used different languages. Future questionnaires evaluating monolingual language experiences should include more detailed questions about passive exposure and passive use of languages so that qualitative information regarding the type of passive use can be evaluated. Taken together, these results show that foreign languages are present in the lives of English-speaking monolinguals, not only in the context of language learning but also through passive exposure. In some cases, this exposure even leads to passive and active use.

Apart from being exposed to foreign languages within the home environment (UK), some monolinguals had also been exposed to foreign languages while abroad. Around half of the participants who had stayed abroad (see Fig 4) had used foreign languages during their visits. These results indicate that the use of foreign languages is frequent among monolinguals who spend time abroad. This is another aspect that should not be overlooked when recruiting or assessing monolingual participants.

Considering that speaking and learning foreign languages is less common in the UK than in other European countries [40], we expected that English-speaking monolinguals from the UK would be the most homogeneous in foreign language exposure and use [44, 45]. Interestingly, the studied English-speaking monolinguals reported instances of foreign language exposure and use. Based on this, we would expect monolinguals from other countries in which foreign language use is more common or monolinguals who are immersed in a more bilingual environment (e.g., monolinguals living in Luxembourg) to show even higher rates of foreign language exposure and use.

## The impact of heterogeneous language experiences on bilinguals' and monolinguals' cognition

Recent meta-analysis and reviews suggest that the effects of bilingualism on cognition might not be as consistent as once thought (e.g., [17, 18]). In the light of these findings, researchers have been trying to disentangle which specific aspects of bilingualism (e.g., the different communicative contexts in which bilinguals are immersed or the amount and type of language switching) impact other cognitive processes (for a recent review, see [46]). For example, the Adaptive Control Hypothesis [47] suggests that bilinguals engage different cognitive processes depending on the interactional contexts. Following the Adaptive Control Hypothesis, when bilinguals use two languages within one context without mixing them (i.e., dual-language context), they practise, among others, conflict monitoring. Still, when two languages are mixed within the same conversation, it is opportunistic planning that is practised. Therefore, the differences observed in cognitive performance among bilinguals seem to depend on their specific type of language use.

Current research suggests that the impact of bilingualism on other cognitive domains is not always a product of long-term language use and even occurs after short-term language context manipulations in bilinguals (e.g., [7]; for a review, see [9]). Moreover, studies with monolinguals have also shown that shorter-term foreign language learning impacts other cognitive processes in monolingual adults [26] and children [25]. In addition to the impact of foreign language learning, passive language exposure can also impact other cognitive processes, as shown in the studies by Bice and Kroll [21] or Kurkela et al. [23]. These studies show that even short-term language exposure leaves traces that cannot be easily ignored. Taking into consideration the results of previous studies and our own findings, we conclude that the language

experiences of individuals who identify themselves as monolingual should be considered in future studies because some monolinguals might also show enhanced performance in tasks that require the exertion of monitoring or executive control, as bilinguals do (e.g., [8])

To sum up, the available evidence indicates that passive exposure and use of foreign languages in monolinguals have crucial consequences for language and cognition (e.g., [21–26, 48]). If both long- and short-term exposure to and use of foreign languages have been shown to affect other cognitive processes in bilinguals, then the impact of foreign language exposure and use in monolinguals should not be ignored. Developing tools that evaluate both bilingual and monolingual language experiences would allow researchers to conduct more ecologically valid studies on the impact of the whole spectrum of language experiences on different cognitive processes. Not taking into consideration the exposure to and use of foreign languages among monolingual participants could obscure comparisons between bilinguals and monolinguals in specific tasks (e.g., executive control tasks); in turn, this would play a role in the replicability crisis and the null and mixed results reported in experimental studies and meta-analyses.

Insights for future research. Our study shows that finding "pure monolinguals" might not be easy or even doable in practical terms. In fact, only 8.63% of our participants (that is, 83 out of 962 participants) reported no learning of foreign languages, no exposure to foreign languages in the UK, passive use of only English, and no stays in non-English-speaking countries. Therefore, studies evaluating the effects of monolingualism versus bilingualism on language and cognition should follow a more ecological approach and consider language experiences as a continuum: from "pure" monolingualism to "extreme" multilingualism (i.e., native-like proficiency and equal exposure and use of all the languages).

Although some researchers have used questionnaires that measure bilingual experiences in order to categorize participants as monolinguals or bilinguals (e.g., the LEAP-Q in studies [49, 50]) or to position all participants on a continuum (e.g., the LSBQ in studies [39, 51–54]), we believe that such questionnaires, in their current format, do not capture the particularity of the monolingual language experience. We propose that questionnaires originally developed for bilinguals could be adapted to increase the specificity of the questions and to better capture the language experiences of people who consider themselves monolingual. Such modifications would subsequently improve both the categorization of participants as monolinguals or bilinguals and the evaluation of their language experiences. For example, the LSBQ [35] contains a detailed assessment of language use; nevertheless, it instructs participants to include languages that they can "speak and understand" ("List all the languages and dialects you can speak and understand including English, in order of fluency"). In this way, many self-categorized monolinguals (but even some bilinguals) might not report other languages to which they have been passively exposed or which they use sporadically. The LEAP-Q [37], another widely used questionnaire, includes several questions on current active and passive use; however, these questions refer to the languages that participants "know" (e.g., "Please list all the languages you know in order of dominance", "Please list all the languages you know in order of acquisition (your native language first)"), again excluding language experiences that do not depend on high proficiency (e.g., short exposure, sporadic use). In addition, the terms "know", "speak", or "understand" are very subjective, and two people with similar knowledge of a language might provide a completely different assessment. This is particularly relevant when evaluating individuals who categorize themselves as monolinguals since, by default, they might complete the questionnaire with the preconception that they "know", "speak", or "understand" only their native language. Taking this into consideration, we propose that a more general term, such as "learn", could be included in the LSBQ [35] and the LEAP-Q [37], as learning does not necessarily imply knowing (e.g., for the LSBQ [35]: "List all the languages and dialects you can

speak, understand, or that you have ever learned, including English, in order of fluency"; for the LEAP-Q [37]: "Please list all the languages you know or have ever learned in order of dominance", "Please list all the languages you know or have ever learned in order of acquisition (your native language first)"). This would allow for more responses to the questions from both monolinguals and bilinguals, as they might be willing to report their experiences with languages that they have learned or encountered at some point in their lives but which they do not necessarily "speak", "understand", or "know". As a consequence, the specificity and heterogeneity of the language background information collected would also improve.

Apart from adjusting existing questions, questionnaires evaluating language experiences also take into consideration different forms of passive exposure. This can be done by including questions on passive language exposure, which does not require interacting with a foreign language or using it. In addition, these questions should evaluate passive exposure across different contexts (e.g., exposure at home vs. at work vs. in other situations). While the LEAP-Q [37] already asks participants to report their level of "current exposure" to their languages in different situations, this question is only given to participants who previously indicated that they "know" a language. In addition, all exposure settings that participants can select on the LEAP-Q [37] involve some kind of language use (either passive use, such as watching TV, or active use, such as interacting with friends or family), which means that the current way in which the question is formulated actually overlooks passive exposure completely. Therefore, apart from measuring active and passive language use, it is crucial to include questions on passive exposure that really measure passive exposure, without merging exposure and use.

Questionnaires including separate items for passive exposure and for passive use should clearly differentiate them, as these two terms might be confusing for the responders. Passive exposure focuses on situations in which participants encounter foreign languages in their environment without interacting with them (e.g., English monolinguals living in the UK who have Spanish speakers as neighbours and hear them talking in Spanish in the vicinity; English monolinguals living in Ireland who frequently hear people taking in Irish). On the other hand, passive use focuses on situations in which participants intentionally engage in activities that involve foreign languages, and these activities require reading (e.g., reading a supermarket leaflet) or listening (e.g., listening to radio advertisements). Monolinguals might not passively or actively use foreign languages, but they might be passively exposed to them on a daily basis as passive exposure does not require interacting purposefully with these languages. This could be the case, for example, if they live within a predominantly Spanish or Chinese neighbourhood in the UK. Thus, including questions that separately tap into passive language exposure and passive language use would provide a more in-depth understanding of monolingual linguistic experiences. Moreover, questions on passive exposure would also be beneficial for bilingual participants because they might be passively exposed to other languages in addition to the ones they know. To facilitate the subsequent interpretation of the responses and ensure that participants understand the difference between passive exposure and passive use, we propose giving examples of situations that involve passive exposure and passive use, as well as having participants specify situations in which they have been passively exposed to foreign languages and in which situations they have passively used the languages they report.

Lastly, questionnaires should also measure and compare present and past passive language use in various activities and across different environments. In general, these types of questions might not be so relevant in the case of balanced bilinguals that use different languages daily, but they would be very informative in monolinguals and unbalanced bilinguals who might not have the constant presence of foreign languages in their environment. This can be especially important with regards to short-term changes in people's life circumstances (e.g., short-term stays abroad).

## Limitations of this study

We acknowledge that the methodology used in this study is not free of limitations. First, the survey included only four main questions completed by all participants (i.e., 1. learning foreign languages: yes/no; 2. exposure to foreign languages in the UK: yes/no; 3. living abroad in non-English-speaking countries: yes/no; 4. list of passively used languages and percentage of passive use of each language). Out of these four questions, the first three included a few nested items that were answered only by a subset of participants (e.g., only participants who answered "yes" to the question on learning languages were subsequently asked to report the list of languages they had learned). The limited number of general questions that were answered by all participants might still have resulted in a rather simplistic view of monolingual language experiences. Second, we did not ask participants to indicate the settings in which they were passively exposed to foreign languages or in which they passively used foreign languages, thus reducing the inferences that can be extracted from the data. Third, and related to the previous point, we did not specify what we meant by "language exposure" in order to avoid restricting participants' responses to specific situations. This might have caused different interpretations of this question, which was especially relevant as we explored the difference between passive language exposure and passive language use. For exposure, it is not necessary to interact with a foreign language in any manner or even pay attention to it, while intentional involvement with a foreign language is required for its passive use. Indeed, participants reported more passive exposure to foreign languages (40%) than passive use of foreign languages (26%). However, it is critical that future studies include specific statements on what passive exposure and passive language use mean, and examples should be given of specific situations that can entail passive exposure and passive use.

One last limitation to consider is that we did not ask participants to complete any other existing linguistic questionnaire apart from our survey, as we aimed to explore specific experiences not measured with classic questionnaires. Future studies could compare participants' responses to the survey presented in this paper (or to another survey that aims to explore monolingual experiences) with responses to existing questionnaires developed for bilinguals but that have been previously used with monolinguals (e.g., LSBQ [35]). This would allow, for example, the comparison of participants' responses when they are prompted with words such as "speak", "understand", or "know" versus when they are also prompted with more general terms, such as "learn". These comparisons would provide better insights on how the available questionnaires could be adapted to gather as much (relevant) information as possible. To conclude, this study shows the importance of further investigation into understanding linguistic experiences in monolinguals and bilinguals and gives suggestions on how to move the field forward.

## Conclusions

This is the first study that evaluates the language experiences of a large group of native English-speaking monolinguals from the UK. Our results indicate that individuals who identify themselves as monolinguals not only learn foreign languages but are also frequently exposed to them (both in the UK and abroad). Sometimes, this exposure also results in active language use (e.g., one fourth of participants who had been exposed to foreign languages in the UK reported using them). Therefore, the language experiences of individuals who declare being monolingual should be considered not as "pure" and homogeneous but as heterogeneous. There is clearly a need for more research that aims to describe and evaluate the heterogeneity of the monolingual language experience, as well as the development and/or adaptation of (existing) questionnaires that evaluate the nuances of the "monolingual" language

experience in detail. For example, not only asking whether foreign languages are being used but specifying whether this use is passive (e.g., watching TV) or active (e.g., conversation), and including questions on passive language exposure. In turn, this information can then be considered in research on the impact of foreign language experiences on cognitive processes.

## Supporting information

**S1 Appendix. Pre-screening filters (Prolific).**
(DOCX)

**S2 Appendix. Complete survey.**
(DOCX)

**S1 Table. List of languages/dialects/types of jargon learned by participants.** Number of participants is shown between brackets.
(DOCX)

**S2 Table. List of foreign languages/dialects to which participants have been exposed in the UK.**
(DOCX)

**S3 Table. List of languages/dialects passively used by participants.** Number of participants is shown between brackets.
(DOCX)

**S4 Table. Non-English-speaking countries reported as temporary countries of residence.** Number of participants is shown between brackets.
(DOCX)

## Acknowledgments

The authors would like to thank all members of our Psychology of Language and Bilingualism Laboratory (LangUsta) for their contributions to the design of the survey. We also extend our gratitude to the research assistant Piotr Rutkowski for his help in data coding, to Michał Remiszewski for his administrative support, and to Michael Timberlake for proofreading.

## Author Contributions

**Conceptualization:** Sofía Castro, Zofia Wodniecka, Kalinka Timmer.

**Data curation:** Sofía Castro.

**Formal analysis:** Sofía Castro.

**Funding acquisition:** Zofia Wodniecka.

**Investigation:** Sofía Castro.

**Methodology:** Sofía Castro, Zofia Wodniecka, Kalinka Timmer.

**Project administration:** Sofía Castro, Zofia Wodniecka.

**Resources:** Zofia Wodniecka.

**Software:** Sofía Castro.

**Supervision:** Zofia Wodniecka, Kalinka Timmer.

**Visualization:** Sofía Castro.

**Writing – original draft:** Sofía Castro, Kalinka Timmer.

**Writing – review & editing:** Sofía Castro, Zofia Wodniecka, Kalinka Timmer.

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
