## [Decision Letter · Decision Letter 0]

20 Sep 2021

PONE-D-21-22659Am I truly monolingual? The importance of understanding linguistic experiences in monolingualsPLOS ONE

Dear Dr. Castro,

Thank you for submitting your manuscript to PLOS ONE. After careful consideration, we feel that it has merit but does not fully meet PLOS ONE’s publication criteria as it currently stands. Therefore, we invite you to submit a revised version of the manuscript that addresses the points raised during the review process.

Both Reviewers and I recognize the value of your work and find the issue under investigation timely and relevant. Said that, as you will see in the comments below, the two Reviewers were not equally enthusiastic of the way the issue was addressed. Reviewer 1 asked for clarification on the need for questionnaires specific for monolinguals. Reviewer 2 has raised serious concerns on the tool you used in the study, and in the way some constructs are operationalized. My own reading of the manuscript is more close to Reviewer 2, and for this reason my decision is for a Major Revision. Note that the invitation to revise the paper is not a guarantee of a final positive outcome - I will send the paper out for a second round of revision, and I'll base my decision on the outcomes of this second round. Thus, when revising the manuscript, please carefully consider all the points raised by both Reviewers.

We look forward to receiving your revised manuscript.

Kind regards,

Simone Sulpizio

Academic Editor

PLOS ONE

Journal Requirements:

2. Please note that according to our submission guidelines (http://journals.plos.org/plosone/s/submission-guidelines), outmoded terms and potentially stigmatizing labels should be changed to more current, acceptable terminology. To this effect, please replace 'Caucasian' with 'white' or 'of European descent' (as appropriate).

Reviewers' comments:

Reviewer's Responses to Questions

**Comments to the Author**

1. Is the manuscript technically sound, and do the data support the conclusions?

Reviewer #1: Yes

Reviewer #2: No

2. Has the statistical analysis been performed appropriately and rigorously? 

Reviewer #1: Yes

Reviewer #2: No

3. Have the authors made all data underlying the findings in their manuscript fully available?

Reviewer #1: Yes

Reviewer #2: Yes

4. Is the manuscript presented in an intelligible fashion and written in standard English?

Reviewer #1: Yes

Reviewer #2: Yes

5. Review Comments to the Author

Reviewer #1: The present study examines the linguistic experiences of native English speakers in the UK who would typically be classified as monolinguals. A questionnaire detailing aspects of active and passive non-English language use was administered to a large group of native-English speaking participants who were born- and living in the UK at time of participation. The data from the questionnaire indicate most participants had some degree of exposure to a foreign language, although degree and nature varied. The authors conclude that such experiences may not be captured by more widely used ‘bilingual’ questionnaires and thus participants considered monolingual should fill out separate questionnaires which tap into more specific aspects of their language use (e.g., passive exposure).

The paper is very interesting and adds to a growing body of studies suggesting most ‘functional’ monolinguals are not truly monolingual, but also have exposure in some sense to non-native languages. The results of this study highlight an important consideration for future research in the field when using ‘monolingual’ participants, particularly studies examining neural or cognitive outcomes of language experience. This said, I have some comments on aspects of the manuscript which I would like to see addressed before recommending for publication.

Comments:

-One of the main points the authors make is that current ‘bilingual’ questionnaires are not sufficient to capture variability in language experience for monolinguals, yet many of the questions included in the survey for the present study (e.g., listening to music; living in a non-native language-speaking country) do exist in some form in many of the bilingual questionnaires. The main point made here is that bilingual questionnaires only prompt responses to language experiences for languages participants know or understand rather than those they either once understood or are only passively exposed to. Given this, one gets the impression that many of these bilingual questionnaires could just be updated to reflect these changes in exposure. I would like to see the authors could expand on why a separate monolingual questionnaire is needed rather than the bilingual questionnaires be updated.

-Alternatively, if a separate questionnaire is really needed to capture variability in language experience more accurately/specifically for monolinguals, it would be nice to see some further discussion of future directions on the goal of a monolingual questionnaire. For example, how would data from monolingual questionnaires be used in comparisons to data from bilingual questionnaires in research examining effects of language experience on cognitive processes?

-On a related note, it would be useful to know if the authors could further defend why they wouldn’t did not also a bilingual questionnaire such as the ones the authors highlight in the introduction. Note that I am not advocating that they should have sent a bilingual questionnaire to the same participants, rather that a bilingual questionnaire could have also been collected from monolinguals in the same environment to see how trends in each compared.

Minor comments:

p 9 line 190: “Five participants were removed due to technical errors…” The way this is worded seems a little confusing. I would suggest re-wording to something like “Five participants were initially removed from the dataset for the following reasons: technical errors (1 participant), …”

Reviewer #2: This manuscript attempts to address a very important topic in the field of bilingual language research, mainly that even the concept of monolingualism is under scrutiny. However, as I point out in my review in more details, the current way in which this very important question has been addressed might not have used a survey tool that is sufficiently sophisticated to address this research question. Below I specify some of the major points.

MAJOR COMMENTS:

-For how interesting the analysis on COVID 19 is, this manuscript is not focused on the changes in language use due to the pandemic, and the results in either case are not embedded in a broader theory. I would suggest taking the section on COVID-19 out.

-The concept of Passive use is not well defined in the manuscript, and could likely also be misleading. For example: I read in the full questionnaire appendix: “Think about your passive use of languages (i.e., watching movies or series, browsing on the internet, listening to music or radio).” In what way is this passive use of a language? I would rather say that this is using a language primarily for comprehension and not production. The authors should better justify why they mean by “passive language use”.

-In a similar vein: regarding the concept of “exposure”. While reading the survey prompt it is not very clear what “exposure” might mean in this context. It might be clearer to experts in the field, but my worry is that when participants completed the survey (if I understand online, and thus with no opportunity to double check with the experimenter) they might not have gotten what a clear definition of “exposure is”. Does it mean interacting (even passively) with one or more individuals who speak another language? Does it mean hearing consistently another language in the work or school, or family environment? This is particularly relevant given the large amount of languages that have been reported (See Table 4). Here it would be imperative to understand the nature of this “exposure”. Unfortunately, I am not convinced that the prompt was detailed enough to clarify what exposure is but also to ask the nature of that exposure. The authors themselves agree in the general discussion for the concept of Passive use that :” Unfortunately, participants were not asked to specify during which activities they passively used these languages. Future questionnaires…”. This is true also for the construct of “exposure”.

-Altogether, I do realize that this was a first attempt to ask a number of questions on monolingual language variability, but I do have the feeling that the survey that has been used is vastly underspecified and does not enable to analyze ad hoc “constructs” or “factors” that are the key of investigation. In other words, if the authors would have created the survey based on established survey methods that for example would enable the creation of specific questions geared towards measuring behavior for specific constructs “such as using a different language passively” -defining as per what I mentioned above what passive means-, they could have had a better lens to understand the phenomena they are trying to analyze.

-I also feel that the data that has been collected here the statistical analyses performed are highly simplistic. I do not want to take away anything from descriptive statistics, but I would encourage the authors to think about more sophisticated ways in which they could analyze the data attempting for example to use a factor analysis to describe what factors seem to be predominant to explain variability in language use?

MINOR:

-Was the survey created based on other used LHQs? If so, please specify

Page 4: Typo Finnish

6. PLOS authors have the option to publish the peer review history of their article (what does this mean?). If published, this will include your full peer review and any attached files.

Reviewer #1: No

Reviewer #2: No

---

## [Author Response · Author response to Decision Letter 0]

28 Oct 2021

Thank you for submitting your manuscript to PLOS ONE. After careful consideration, we feel that it has merit but does not fully meet PLOS ONE’s publication criteria as it currently stands. Therefore, we invite you to submit a revised version of the manuscript that addresses the points raised during the review process.

Both Reviewers and I recognize the value of your work and find the issue under investigation timely and relevant. Said that, as you will see in the comments below, the two Reviewers were not equally enthusiastic of the way the issue was addressed. Reviewer 1 asked for clarification on the need for questionnaires specific for monolinguals. Reviewer 2 has raised serious concerns on the tool you used in the study, and in the way some constructs are operationalized. My own reading of the manuscript is more close to Reviewer 2, and for this reason my decision is for a Major Revision. Note that the invitation to revise the paper is not a guarantee of a final positive outcome - I will send the paper out for a second round of revision, and I'll base my decision on the outcomes of this second round. Thus, when revising the manuscript, please carefully consider all the points raised by both Reviewers.

Reviewer #1:

The present study examines the linguistic experiences of native English speakers in the UK who would typically be classified as monolinguals. A questionnaire detailing aspects of active and passive non-English language use was administered to a large group of native-English speaking participants who were born- and living in the UK at time of participation. The data from the questionnaire indicate most participants had some degree of exposure to a foreign language, although degree and nature varied. The authors conclude that such experiences may not be captured by more widely used ‘bilingual’ questionnaires and thus participants considered monolingual should fill out separate questionnaires which tap into more specific aspects of their language use (e.g., passive exposure).

The paper is very interesting and adds to a growing body of studies suggesting most ‘functional’ monolinguals are not truly monolingual, but also have exposure in some sense to non-native languages. The results of this study highlight an important consideration for future research in the field when using ‘monolingual’ participants, particularly studies examining neural or cognitive outcomes of language experience. This said, I have some comments on aspects of the manuscript which I would like to see addressed before recommending for publication.

Comments:

1. One of the main points the authors make is that current ‘bilingual’ questionnaires are not sufficient to capture variability in language experience for monolinguals, yet many of the questions included in the survey for the present study (e.g., listening to music; living in a non-native language-speaking country) do exist in some form in many of the bilingual questionnaires. The main point made here is that bilingual questionnaires only prompt responses to language experiences for languages participants know or understand rather than those they either once understood or are only passively exposed to. Given this, one gets the impression that many of these bilingual questionnaires could just be updated to reflect these changes in exposure. I would like to see the authors could expand on why a separate monolingual questionnaire is needed rather than the bilingual questionnaires be updated.

We appreciate this valuable comment as we agree with the Reviewer that adaptations of the existing bilingual questionnaires would indeed allow heterogenous foreign language experiences across individuals to be reflected (both monolinguals and bilinguals). Therefore, we have explored this possibility in the revised version of the manuscript by suggesting ways in which some well-known bilingual questionnaires (i.e., LSBQ, LEAP-Q) could be adapted to measure language exposure and use among individuals who identify as monolingual. Information on this topic can be found on pages 22–24 (lines 486–540 of the clean version). The abstract has also been modified accordingly (page 2, line 44 of the clean version).

2. Alternatively, if a separate questionnaire is really needed to capture variability in language experience more accurately/specifically for monolinguals, it would be nice to see some further discussion of future directions on the goal of a monolingual questionnaire. For example, how would data from monolingual questionnaires be used in comparisons to data from bilingual questionnaires in research examining effects of language experience on cognitive processes?

We thank the Reviewer for the comments and suggestions regarding this topic. As we indicate in response to Comment #1, we agree with the idea that bilingual questionnaires could indeed be adapted to reflect monolingual experiences, and that a specific questionnaire for monolinguals might not be required. In the revised version of the manuscript, we have included some ideas on how bilingual questionnaires could be adapted (see our response to Comment #1), and how developing a questionnaire that is valid for both groups could improve research on the effects of language experiences in other processes (page 22, line 469 of the clean version).

3. On a related note, it would be useful to know if the authors could further defend why they wouldn’t did not also a bilingual questionnaire such as the ones the authors highlight in the introduction. Note that I am not advocating that they should have sent a bilingual questionnaire to the same participants, rather that a bilingual questionnaire could have also been collected from monolinguals in the same environment to see how trends in each compared.

We agree with the Reviewer that including a bilingual questionnaire would have increased the validity of our findings. In fact, even asking the same group of participants to complete both questionnaires would have been interesting as it would have allowed us to compare, e.g., the languages reported by participants when they are prompted with words like ‘speak’, ‘understand’, or ‘know’ (as in LSBQ or LEAP-Q) versus when they are prompted with the words ‘learn’ or ‘exposed’ (as in our questionnaire). The reason for not including a bilingual questionnaire was that we wanted to focus on experiences that might be specific for monolinguals and which, in our opinion, are not assessed by traditional questionnaires. However, we have included a comment about the possibility of this comparison in the “Limitations of this study” section (page 26, line 563 of the clean version).

Minor comments:

1. p 9 line 190: “Five participants were removed due to technical errors…” The way this is worded seems a little confusing. I would suggest re-wording to something like “Five participants were initially removed from the dataset for the following reasons: technical errors (1 participant), …”

We understand and apologize for the confusion caused by our original statement. We have modified the sentence accordingly (page 9, line 214 of the clean version).

Reviewer #2:

This manuscript attempts to address a very important topic in the field of bilingual language research, mainly that even the concept of monolingualism is under scrutiny. However, as I point out in my review in more details, the current way in which this very important question has been addressed might not have used a survey tool that is sufficiently sophisticated to address this research question. Below I specify some of the major points.

MAJOR COMMENTS:

1. For how interesting the analysis on COVID 19 is, this manuscript is not focused on the changes in language use due to the pandemic, and the results in either case are not embedded in a broader theory. I would suggest taking the section on COVID-19 out.

We thank the Reviewer for this comment. When we prepared the first version of this manuscript, given the dynamic situation with the COVID-19 pandemic, we wanted to make sure that the possible changes in passive language use due to this situation were incorporated in the survey. However, since the results indicate no difference in language use before and during the COVID-19 pandemic in the tested sample, we agree that reporting such detailed analysis is indeed not relevant for the purpose of this paper. Therefore, we have removed this analysis from the results and only make a short note about it for the sake of transparency in the methods section (page 10, line 231 of the clean version). 

2. The concept of Passive use is not well defined in the manuscript, and could likely also be misleading. For example: I read in the full questionnaire appendix: “Think about your passive use of languages (i.e., watching movies or series, browsing on the internet, listening to music or radio).” In what way is this passive use of a language? I would rather say that this is using a language primarily for comprehension and not production. The authors should better justify why they mean by “passive language use”.

We thank the Reviewer for this suggestion; it helped us realise that the concept of passive use was not as clearly explained in the paper as we thought it was. We agree with the Reviewer that what we refer to as “passive use” is the same as comprehension. We intentionally did not use the word “comprehension” in the survey to avoid participants associating “comprehension” with “high language proficiency”. Such an association might cause participants to disregard some languages for which they only have minimal receptive use as they consider themselves as “knowing” only one language proficiently (as stated in the Prolific profiles). In addition, although a limited degree of comprehension is needed when, e.g., browsing the internet, this knowledge can be very basic and participants might not consider this type of language use as “comprehending” in the general sense of the word. 

To ensure that the meaning of “passive use” and our choice of words is understandable for our future readers, we have included clarifications across the manuscript: for example, in the Abstract (page 2, line 34 of the clean version), the Introduction (page 6, line 125 of the clean version), and the Methods (page 9, line 197 of the clean version).

3. In a similar vein: regarding the concept of “exposure”. While reading the survey prompt it is not very clear what “exposure” might mean in this context. It might be clearer to experts in the field, but my worry is that when participants completed the survey (if I understand online, and thus with no opportunity to double check with the experimenter) they might not have gotten what a clear definition of “exposure is”. Does it mean interacting (even passively) with one or more individuals who speak another language? Does it mean hearing consistently another language in the work or school, or family environment? This is particularly relevant given the large amount of languages that have been reported (See Table 4). Here it would be imperative to understand the nature of this “exposure”. Unfortunately, I am not convinced that the prompt was detailed enough to clarify what exposure is but also to ask the nature of that exposure. The authors themselves agree in the general discussion for the concept of Passive use that :” Unfortunately, participants were not asked to specify during which activities they passively used these languages. Future questionnaires…”. This is true also for the construct of “exposure”.

We agree with the Reviewer that the way in which we formulated the exposure question might have been too general, thus causing misunderstandings for our participants. Our goal when developing the survey was to avoid restricting participants’ answers to one example. Therefore, we decided to include only a general statement. However, due to the attentive observation of the Reviewer, we realised that we might have been too broad and that a more detailed explanation of what we meant by “exposure” should have been presented to the participants. We have now included this issue as a limitation in the revised version of the manuscript (page 25, line 553 of the clean version).

In addition, and following the second part of the Reviewer’s comment, in this revised manuscript we now mention that the questions on passive exposure may not have been detailed enough (i.e., we did not ask participants to report the environments in which they had been exposed to foreign languages), similarly to what happened with the issue of the passive use of languages. We now discuss the interpretation of these terms on page 20, line 416 of the clean version, and on page 25, line 551 of the clean version.

4. Altogether, I do realize that this was a first attempt to ask a number of questions on monolingual language variability, but I do have the feeling that the survey that has been used is vastly underspecified and does not enable to analyze ad hoc “constructs” or “factors” that are the key of investigation. In other words, if the authors would have created the survey based on established survey methods that for example would enable the creation of specific questions geared towards measuring behavior for specific constructs “such as using a different language passively” -defining as per what I mentioned above what passive means-, they could have had a better lens to understand the phenomena they are trying to analyze.

We agree with the Reviewer that our survey has limitations that affect the scope of possible analyses and the conclusions that can be drawn from this study. As the Reviewer mentions, this study is a first attempt to explore language experiences in monolinguals. We conducted only a short survey with the goal of obtaining a general overview of monolinguals’ foreign language experiences and providing initial guidance for future studies, but we did not aim to validate a new questionnaire or to analyse in-depth the constructs of monolingual language experiences (yet). For us, this initial exploratory step was necessary to raise awareness of the potential caveats related to oversimplification of monolingual language experiences and to initiate discussion on this issue among researchers. Based on the findings obtained, the next step will be to develop fine-grained questionnaires or adapt existing bilingual questionnaires to explore different constructs that may characterise the monolingual language experience (e.g., “passive language exposure”, “foreign language learning at school”). 

We agree that some aspects of the survey could have been developed differently (e.g., increasing the number of items, having more than one item per construct, better specification of target concepts such as language exposure, etc.), and more fine-grained questions should be used in future studies. We have included a paragraph in the “Limitations of this study” section, in which we discuss all the limitations and provide solutions concerning how they could be addressed in follow-up studies (page 25, line 542 of the clean version).

In addition, we have also highlighted the preliminary nature of this study on page 4 (line 75 of the clean version), and on page 6 (line 137 of the clean version).

5. I also feel that the data that has been collected here the statistical analyses performed are highly simplistic. I do not want to take away anything from descriptive statistics, but I would encourage the authors to think about more sophisticated ways in which they could analyze the data attempting for example to use a factor analysis to describe what factors seem to be predominant to explain variability in language use?

We appreciate the Reviewer’s suggestion that more sophisticated analyses should be performed. We did consider the possibility of running other types of analyses while preparing the initial submission; however, after having discussed it in our team and after consulting statistical manuals, we decided to report only descriptive information as our survey includes a small number of items. 

To obtain meaningful results from a factor analysis, it is recommended to have at least three items loading onto each factor (e.g., three items on language learning) (Hair et al., 2010). However, we had only four main items that were answered by all participants (i.e., if participants learned languages, if they lived abroad, if they were exposed to foreign languages in the UK, and the number of languages they passively used together with the percentage of time they passively used each language). The other items (e.g., language proficiency, number of stays abroad, number of languages to which participants were exposed in the UK, etc.) were nested questions, meaning that only participants who answered ‘yes’ to the main questions were prompted to answer the nested questions. Due to the large number of missing data points that we would get from nested questions, only the main four questions could be included in the factor analysis, and such a small number of items would not be enough to obtain even two meaningful factors.

Nevertheless, following the Reviewer’s suggestion and to test this approach, we ran a two-factor solution including our four main items (learning foreign languages – yes/no; exposure to foreign languages – yes/no; living abroad – yes/no; number of languages passively used by participants – 1/2/3/4). We chose a two-factor solution for practical reasons: it would not be interesting to explore a unifactorial solution as we would expect monolingual language experience to be comprised of different constructs; however, with the limited number of items we had, testing larger factor solutions would have been methodologically impossible. We ran the factor analysis in R using the “psych” package (version 2.1.9; Revelle, 2021) with maximum likelihood method and oblimin rotation so that the two factors could correlate. The results showed that passive exposure to foreign languages and passive use of foreign languages loaded onto one factor (factor loadings 0.50 and 0.62, respectively; eigenvalue = 1.41) and that learning foreign languages loaded onto the other factor (factor loading = 0.56; eigenvalue = 1.08). Living abroad did not load onto either of these factors (factor loadings < 0.40). These results might suggest that both passive exposure/passive use of foreign languages and learning foreign languages are relevant when defining monolingual language experiences, with the former having more weight than the latter. As interesting as these results would be with a larger questionnaire, we honestly believe that our methodological limitations prevent us from including this analysis in the revised version of manuscript as we do not have enough items loading onto each factor.

Another analysis that we considered while preparing the first version of this manuscript was goodness-of-fit chi-square tests with the following items:

- Item 1: “Have you ever used any of the language(s)/dialect(s) that you learned for any reason?” Yes/No

- Item 2: “Have you ever used any of the language(s)/dialect(s) to which you have been exposed in the UK for any reason?” Yes/No

- Item 3: “While living abroad, did you use any other language(s)/dialect(s) apart from your native language to communicate?” Yes/No

We thought about testing a 0 (answer ‘yes’) /100 (answer ‘no’) distribution. The logic of our reasoning was that although, e.g., learning languages and being exposed to foreign languages might be inevitable in modern societies, monolinguals would not report any foreign language use in such circumstances (i.e., they should answer ‘no’ to these three items). We decided not to include such analyses in the original manuscript because this hypothesis was not clearly established a priori and we did not have any previous data or scientific literature to guide us. We indeed did not expect to find such a large number of monolinguals reporting ‘use’ of foreign languages, but we had not established any expected distribution before analysing the data. In addition, considering our initial goal (i.e., to explore different monolingual language experiences), we considered that the inclusion of these results would not add to our study. 

However, following your comment, we decided to run these goodness-of-fit chi-square tests. The tests were run in R, and we tested a 95 (answer ‘no’) / 5 (answer ‘yes’) distribution for each of the items. We could not test a 0/100 distribution because the expected values for one of the conditions (in our case, answer ‘yes’) would be below 5, and this might cause the chi-square approximation to be incorrect. The results from the chi-square tests showed that the distribution of our three items statistically differed from a 95 (answer ‘no’) /5 (answer ‘yes’) distribution:

- Item 1 (using languages learned): X2 (1, N = 801) = 3955.70, p < .001. 

- Item 2 (using languages to which participants were exposed): X2 (1, N = 376) = 300.01, p < .001. 

- Item 3 (using foreign languages abroad): X2 (1, N = 89) =534.83, p < .001. As the tested distribution for this item still included less than 5 observations in the expected sample size for the answer ‘yes’, the chi-square approximation should be interpreted with caution. To overcome this problem, we repeated the analysis using Monte-Carlo simulations for the p-values (10000 replicates), and the results still indicated that our distribution differed from the tested 95/5 distribution, X2 (NA, N = 89) =534.83, p < .001. The absence of degrees of freedom in this second test is due to the inclusion of Monte-Carlo simulations.

For the reasons stated above, we do not believe that including the chi-square analysis would be appropriate. Nevertheless, if the Reviewer considers that any of these analyses should be reported, we will be happy to include them in the manuscript. To show our awareness of the limited analyses that we could conduct, we have included a paragraph in the “Limitations of this study” section, in which we discuss the reduced number of items included in the survey and how this affects the conclusions that can be inferred (page. 25, line 542 of the clean version).

We hope that our clarifications regarding our analysis approach have reduced some of the Reviewer’s concerns and that the explanations we have provided are sufficiently clear.

MINOR:

1. Was the survey created based on other used LHQs? If so, please specify

The survey was not directly based on any existing LHQ as we wanted to explore specific aspects of the monolingual language experience that are not properly evaluated in the classical bilingual questionnaires (e.g., language exposure). However, the survey was based on the standard procedures used in our laboratory to evaluate language experiences. In addition, the questions about language proficiency did follow the widely used approaches in which the four classical domains of language proficiency are rated (reading, listening, writing, speaking) and the age of acquisition is reported. We have included a clarification about the creation of our survey in the methods section (page 8, line 181 of the clean version). 

2. Page 4: Typo Finnish

Thank you for pointing out this typo; it has been corrected (page 4, line 83 of the clean version).

---

## [Decision Letter · Decision Letter 1]

7 Feb 2022

PONE-D-21-22659R1Am I truly monolingual? Exploring foreign language experiences in monolingualsPLOS ONE

Dear Dr. Castro,

Thank you for submitting your manuscript to PLOS ONE. After careful consideration, we feel that it has merit but does not fully meet PLOS ONE’s publication criteria as it currently stands. Therefore, we invite you to submit a revised version of the manuscript that addresses the points raised during the review process.

I'm sorry for the long time you waited for this decision. I hoped to receive a second opinion before taking a decision, but as it was taking too long, I decided to base my decision on one reviews plus my own reading. I agree with the Reviewer that you did a good job in addressing the raised concerns. I appreciated that you clearly acknowledged the limitations of your study, which still I think may be of interest for the bilingualism community. Before recommending publication I ask you to address the new minor issues raised by the Reviewer. 

We look forward to receiving your revised manuscript.

Kind regards,

Simone Sulpizio

Academic Editor

PLOS ONE

Journal Requirements:

Reviewers' comments:

Reviewer's Responses to Questions

**Comments to the Author**

1. If the authors have adequately addressed your comments raised in a previous round of review and you feel that this manuscript is now acceptable for publication, you may indicate that here to bypass the “Comments to the Author” section, enter your conflict of interest statement in the “Confidential to Editor” section, and submit your "Accept" recommendation.

Reviewer #1: (No Response)

2. Is the manuscript technically sound, and do the data support the conclusions?

Reviewer #1: Yes

3. Has the statistical analysis been performed appropriately and rigorously? 

Reviewer #1: N/A

4. Have the authors made all data underlying the findings in their manuscript fully available?

Reviewer #1: Yes

5. Is the manuscript presented in an intelligible fashion and written in standard English?

Reviewer #1: Yes

6. Review Comments to the Author

Reviewer #1: This is a revised version of a previously submitted manuscript examining variability in language experience in monolingual (native English) speakers. The authors have done a good job of addressing the comments from the previous version of the manuscript, and the current version is certainly improved. In reading the manuscript again, I have a couple further comments which I would like to see addressed but am otherwise happy to recommend for publication.

Comments:

-In the ‘Insights for Future Research’ section, it would be useful if the authors could expand on their calls to include separate questions passive exposure in the current bilingual questionnaires. If these are included as a separate section or set of questions, would these questions directly overlap directly with questions pertaining to active use? That is, would the same scenarios be examined, just with passive use?

-On a similar note, I am not sure I agree with the authors calls to removing terms like ‘speak’ or ‘know’ outright from the existing questionnaires in detailing language use, as opposed to adding ‘learned’ to this list. It would be useful if the authors could clarify this a bit further.

Minor comments:

Table 3- the country column is a little confusingly worded to me. I would recommend changing this for clarity to “1 Foreign country, 2 foreign countries…” etc. The same applies to Table 2. I would reword these as “1 foreign language, 2 foreign languages…” etc.

7. PLOS authors have the option to publish the peer review history of their article (what does this mean?). If published, this will include your full peer review and any attached files.

Reviewer #1: No

---

## [Author Response · Author response to Decision Letter 1]

22 Feb 2022

Response to Reviewers

We appreciate the comments and suggestions provided by the Reviewer. All points have been addressed in the revised version of the manuscript. In addition, in the revised version, the reference number 54 has been updated from a preprint to a published article (Voits et al., 2022; page 23, line 504 and page 32, line 760 of the clean version). We have also included three new references in the revised version (numbers 27, 28, 29). The in-text citations for these references are on page 4, line 90 (Byers-Heinlein et al., 2017) and line 92 (Oh et al., 2020; Tobin et al., 2017) of the clean version.

Reviewer #1: 

This is a revised version of a previously submitted manuscript examining variability in language experience in monolingual (native English) speakers. The authors have done a good job of addressing the comments from the previous version of the manuscript, and the current version is certainly improved. In reading the manuscript again, I have a couple further comments which I would like to see addressed but am otherwise happy to recommend for publication.

Comments:

1. In the ‘Insights for Future Research’ section, it would be useful if the authors could expand on their calls to include separate questions passive exposure in the current bilingual questionnaires. If these are included as a separate section or set of questions, would these questions directly overlap directly with questions pertaining to active use? That is, would the same scenarios be examined, just with passive use?

If we understand correctly, the Reviewer is asking us to clarify the difference between passive exposure and passive use. Our suggestion to include specific questions on passive exposure in addition to passive use reflects our interest in obtaining information on whether participants frequently encounter foreign languages without interacting with them or without even paying attention to them (e.g., people who have foreigners as neighbours and hear them talking in their native language in the vicinity; people who have foreign co-workers who speak in their native language at work). On the other hand, questions on passive use would provide information on whether participants intentionally engage in activities that involve reading or listening in a foreign language. Examples of passive use would be reading a supermarket leaflet in a foreign language or listening to radio advertisements in a foreign language. As passive exposure does not require interacting with or using a language, it is possible to be exposed to foreign languages whilst not passively (or actively) using them.

Including separate questions for passive exposure and passive use would provide a more detailed picture of the languages that a participant encounters. In addition, in the case of monolinguals – our target population – passive exposure has been linked to differences in linguistic processing (Bice & Kroll, 2019; Kurkela et al., 2019). Therefore, it is crucial to understand whether monolinguals are passively exposed to foreign languages, as this exposure might have influenced their cognitive processes.

Following the Reviewer’s suggestion and to ensure that both terms are correctly understood and differentiated, we have clarified the difference between passive exposure and passive use the first time these two terms are presented together (page 6, lines 130-136 of the clean version), and we have made an additional clarification in the section “Insights for future research” (page 25, lines 551-566 of the clean version). 

In addition, we agree with the Reviewer that participants might feel an overlap between passive exposure and passive use if the definition is not clear for them. Thus, in addition to defining both terms in the questionnaire, we also propose the inclusion of open-ended questions in which participants could specify in which situations they are passively exposed to foreign languages and in which situations they passively use foreign languages (page 25, lines 566-570 of the clean version)

2. On a similar note, I am not sure I agree with the authors calls to removing terms like ‘speak’ or ‘know’ outright from the existing questionnaires in detailing language use, as opposed to adding ‘learned’ to this list. It would be useful if the authors could clarify this a bit further.

We agree with the Reviewer that completely removing the terms “speak”, “know”, or “understand” is not necessary, and that the word “learn” could be added instead. The inclusion of all these different terms would allow monolinguals and bilinguals to report all their languages without restricting their responses to the languages they know/speak/understand or to the languages they have learned. In addition, including all terms would facilitate using the same survey for monolinguals and bilinguals. We have made some modifications to the revised version of the manuscript to propose the inclusion of all these terms in the questionnaires (page 24, lines 524-535).

Minor comments:

1. Table 3- the country column is a little confusingly worded to me. I would recommend changing this for clarity to “1 Foreign country, 2 foreign countries…” etc. The same applies to Table 2. I would reword these as “1 foreign language, 2 foreign languages…” etc.

Thank you very much for this comment. We have carefully considered your suggestion, but we think that changing the labels to “1 Foreign country, 2 Foreign countries…” could be misleading for future readers. The reason for this is that each row does not exclusively include participants who have lived in only one (or two, or three) foreign countries. In other words, data included in “Foreign country 1” contains all participants who have lived in at least one foreign country; data included in “Foreign Country 2” includes all participants who have lived in at least two foreign countries, and so on. In our study, 89 participants had lived at least in one foreign country; 22 had lived in at least two foreign countries; six had lived in at least three foreign countries; and three had lived in four foreign countries. For purposes of clarity, we have included this information as a note below Table 3 (page 17, line 368-374 of the clean version). 

The same logic applies to Table 2: The “Foreign Language 1” row includes all participants who had been exposed to at least one foreign language; the “Foreign Language 2” row includes all participants who had been exposed to at least two foreign languages, and so on. In this case, a total of 376 participants had been exposed to at least one foreign language; 89 participants had been exposed to at least two foreign languages; and 29 participants had been exposed to three foreign languages. As with Table 3, we have included this information as a note below Table 2 (page 15, lines 326-331 of the clean version). In addition, we have provided some information on page 11 (lines 257-261 of the clean version) on how the countries and languages were categorized. We hope that these changes facilitate the understanding of the table and our data.

---

## [Editor Report · Decision Letter 2]

4 Mar 2022

Am I truly monolingual? Exploring foreign language experiences in monolinguals

PONE-D-21-22659R2

Dear Dr. Castro,

We’re pleased to inform you that your manuscript has been judged scientifically suitable for publication and will be formally accepted for publication once it meets all outstanding technical requirements.

Kind regards,

Simone Sulpizio

Academic Editor

PLOS ONE
---

## [Editor Report · Acceptance letter]

11 Mar 2022

PONE-D-21-22659R2 

Am I truly monolingual? Exploring foreign language experiences in monolinguals. 

Dear Dr. Castro:

I'm pleased to inform you that your manuscript has been deemed suitable for publication in PLOS ONE. Congratulations! Your manuscript is now with our production department. 

Kind regards, 

on behalf of

Dr. Simone Sulpizio 

Academic Editor

PLOS ONE